# Transcription profiles reveal sugar and hormone signaling pathways mediating tree branch architecture in apple (*Malus domestica* Borkh.) grafted on different rootstocks

Yanhui Chen, Xiuhong An, Deying Zhao, Enmao Li, Renpeng Ma, Zhuang Li, Cungang Cheng[ID]*

Key Laboratory of Mineral Nutrition and Efficient Fertilization for Deciduous Fruits, Liaoning Province, Key Laboratory of Fruit Germplasm Resources Utilization, Ministry of Agriculture, Institute of Pomology, Chinese Academy of Agricultrual Sciences, Xingcheng, Liaoning, P. R. China

* ccungang@163.com

**Data Availability Statement:** All relevant data are within the manuscript and its Supporting Information files.

## Abstract

Apple trees grafted on different rootstock types, including vigorous rootstock (VR), dwarfing interstock (DIR), and dwarfing self-rootstock (DSR), are widely planted in production, but the molecular determinants of tree branch architecture growth regulation induced by rootstocks are still not well known. In this study, the branch growth phenotypes of three combinations of 'Fuji' apple trees grafted on different rootstocks (VR: *Malus baccata*; DIR: *Malus baccata/T337*; DSR: *T337*) were investigated. The VR trees presented the biggest branch architecture. The results showed that the sugar content, sugar metabolism-related enzyme activities, and hormone content all presented obvious differences in the tender leaves and buds of apple trees grafted on these rootstocks. Transcriptomic profiles of the tender leaves adjacent to the top buds allowed us to identify genes that were potentially involved in signaling pathways that mediate the regulatory mechanisms underlying growth differences. In total, 3610 differentially expressed genes (DEGs) were identified through pairwise comparisons. The screened data suggested that sugar metabolism-related genes and complex hormone regulatory networks involved the auxin (IAA), cytokinin (CK), abscisic acid (ABA) and gibberellic acid (GA) pathways, as well as several transcription factors, participated in the complicated growth induction process. Overall, this study provides a framework for analysis of the molecular mechanisms underlying differential tree branch growth of apple trees grafted on different rootstocks.

## Introduction

Apples (*Malus domestica*) are planted worldwide, and many factors affect the development and phytochemical profile of apples, such as water, fertilizer, temperature, light intensity, grafted rootstock, etc. As an important part of fruit trees, rootstocks not only affect the absorption, transportation and utilization of mineral nutrients but also play an important role in regulating the growth and development of aboveground trees and fruit quality [1, 2].

**Funding:** This work was supported by the National Natural Science Foundation of China (No. 31701902; No.31201602) to XA, the Program of Developing the Modern Agricultural Industry Technology System (Apple) (CARS-27) to CC and the funding of CAAS-ASTIP to CC.

**Competing interests:** The authors have declared that no competing interests exist.

For examples, the concentrations K and Mg in the scion leaves of trees on vigorous rootstock (VR) were higher than those on the dwarfing rootstock (DR) [3]. The annual shoot compositions of 'Royal Gala' apple trees grafted on DR (M.9) showed earlier transition to flowering than those of trees on non-dwarfing rootstock/dwarfing interstock combinations (MM.106/M.9; DIR) [4]. The firmness, Vitamin C content, and titratable acid were the highest of 'Red General Fuji' fruit of trees DIR than those fruits on VR or dwarfing self-rootstock (DSR). And fruits from the trees on DR had highest soluble solid and soluble sugar content [5].

Although the effects of different rootstocks on grafting have been well studied, the mechanism leading to this differential development is not clear. With the explosion in second-generation sequencing technology, to obtain insight into potential mechanisms underlying the influence of rootstock on scion growth, transcriptome studies have increasingly been used to decipher the molecular mechanisms that lead to this difference [6–11]. Different rootstocks were found to significantly affect the expression of genes involved in the auxin signal transduction pathway and gibberellic acid (GA) biosynthesis pathway in grafted plants and regulated hormone levels and their signaling pathways [8]. Among them, *ARF1*, *GA2OX1* and other genes play an important role. Comparison of peony petals on different rootstocks showed that the differences in gene expression between them were mainly reflected in starch and sucrose metabolism, cell wall polysaccharide modification, redox activity, and signal transduction processes [12]. Comparative transcriptome approaches were used to investigate the effects of different soil compositions and rootstocks on leaf transcriptomic modulation of *Pinot Noir* and the drought tolerant and susceptible rootstock genotypes in response to water stress [13, 14]. A study on the transcriptomes of apple rootstocks showed that genes of sorbitol dehydrogenase, homeobox-leucine zipper, and hevein-like proteins were at higher expression levels in larger trees, while extensin-like gene was confirmed to be expressed at higher levels in smaller trees. This study illustrated the utility of using rootstock-regulated phenotypes to identify genes associated with horticulturally important traits [15].

Rootstocks are selected based on rooting and grafting capacity and abiotic and biotic stress tolerance and beneficially alter the scion phenotype [1]. Because of cold hardiness, *Malus baccata* is widely used as a VR in cold regions of the world [16]. T337 is good strain of the apple DSR which is also widely suitable for grafting apple varieties [17]. The identification of dwarfing gene (*Dw1*, *Dw2*, and *Dw3*) loci in apple rootstock is contribute to explain the genetic mechanism of dwarfing phenotypes [18, 19]. As the photosynthetic organ, the metabolism of leaves reflects the effects of grafting on tree branch architecture growth [2, 20, 21]. Therefore, to analyze the mechanism of rootstock-induced dwarfing, the most direct choice is to analyze leaf growth status. Furthermore, after flower bloom, the shoot growth kinetics of 'Fuji' apple trees present an 'S' model curve [22]. Shoots stop growing at four weeks after flower bloom, and flower bud induction begins at this stage [23, 24]. Moreover, at this important physiological differentiation stage in apples, the leaves adjacent to the top buds of spurs play a significant role in bud development and branch growth by adjusting sugar content, nitrogen (N) content, carbon (C)/N ratios, sugar metabolism-related enzyme activities and hormone contents [22].

These results explain the possible molecular mechanisms of different species under different rootstocks, but the molecular mechanisms underlying the differences caused by different rootstocks are not clear for 'Fuji' apple trees. In this study, we conducted a transcriptome analysis of the tender leaves adjacent to the top buds of shoots grafted on VR, DIR, and DSR. The DIR we used was a *Malus baccata/T337* combination, which connects vigorous rootstock and dwarfing rootstock. Based on analysis of differentially expressed genes (DEGs) found by pairwise comparison, we explored the regulatory pathways related to tree growth differences induced by rootstocks.

## Materials and methods

### Plant material and sample collection

In the Qiansuo apple demonstration base (Xingcheng, Liaoning province, China), 2-year-old 'Fuji' apple trees grafted on different rootstocks (VR: *Malus baccata*; DIR: *Malus baccata/T337*; DSR: *T337*) were planted in May of 2014. Approximately 50 trees were used for measurement of tree-structure indexes in mid-July 2017. Buds on the top of shoots and tender leaves adjacent to the top buds were collected from these trees at the same time and then separately mixed well with liquid nitrogen to determine sugar contents, sugar metabolism-related enzyme activities, and phytohormone contents. Three biological and technical repetitions were used to measure these physiological indexes. In addition, a portion of the tender leaves were ground to powder for RNA extraction and RNA-seq library construction. The sample collection for the construction of RNA library was three biological repeats. Samples VR-1, -2, and -3 were leaves of apple trees on VR; Samples DIR-1, -2, and -3 were leaves of apple trees on DIR; Samples DSR-1, -2, and -3 were leaves of apple trees on DSR.

### Measurements of sugar content and sugar metabolism-related enzyme activities

At the capping period of spring shoots, 1 g fresh weight of leaves or buds were used for sugar extraction, and the contents were determined using a high-performance liquid chromatography (HPLC) system as described by Rosa et al. [25]. Briefly, after being ground with a mortar and pestle, samples were homogenized with 5 mL 80% ethanol in a water bath for 10 min at 75°C and then centrifuged for 10 min at 5000 g. Supernatants were dried under a stream of hot air, and then, the residue was resuspended in 1 mL distilled water and filtrated through an ion-exchange column. The detailed standard measurement protocols were the same as described in Dobrev and Vankova [26]. Additionally, 0.1 g fresh weight of leaves and buds were used to measure sugar metabolism-related enzyme activities via spectrophotometry with commercial kits (Article Nos. BC0580, BC0600, BC2530, BC2470, and BC0570, Solarbio Science and Technology Co., Ltd, Beijing, China). Three biological replicates were prepared.

### Determination of phytohormone content

The auxin (IAA), cytokinin (CK), gibberellin (GA), and abscisic acid (ABA) contents were determined by HPLC as described by Chen et al. [27]. Briefly, phytohormones from 2 g fresh weight of leaves or buds were extracted in 8 mL of acetonitrile extraction medium (containing 30 μg mL$^{-1}$ sodium diethyldithiocarbamate as an antioxidant) at 4°C for 16 h. The collected supernatant was dried under low pressure. Then, the residues were redissolved in 5 mL of phosphate buffer (0.4 M, pH 8.0). After the pigment and hydroxybenzene were removed with trichloromethane and PVPP, phytohormones were extracted twice with ethyl acetate. The ethyl acetate phase was dried under low pressure and redissolved in 1 mL mobile phase buffer. The extractions were separated on a C18 column (4.6 mm × 150 mm, 5 μm, Agilent, Santa Clara, California, United States). Authentic phytohormone standards (Sigma, Shanghai, China) were used to establish standard curves.

### RNA extraction, library construction, and transcriptome sequencing

Total RNA extracted from each tender leaves using Trizol reagent (Invitrogen, Carlsbad, CA, USA) was used for the cDNA libraries and RNA-seq at Novogene Bioinformatics Technology Co. Ltd. (Beijing, China). NEBNext® Ultra™ RNA Library Prep Kit for Illumina® (NEB, USA) was used to generate sequencing libraries according to the manufacturer's instructions,

and HTSeq v0.6.1 was used to calculate the number of reads corresponding to each gene. Fragments per kilobase of exon model per million mapped reads (FPKM) of each gene were calculated based on the length of the gene and the number of reads mapped to the gene [28]. The DESeq R package (1.18.0) was used to perform differential expression analysis of pairs of groups, and the Benjamini and Hochberg approach was used to control the false discovery rate through adjustes P-values. Genes selected out by DESeq with an adjusted P-value < 0.05 were assigned as differentially expressed [29]. GOseq R package was used to implement the gene ontology (GO) enrichment analysis of DEGs [30]. The statistical enrichment of DEGs in KEGG pathways was tested by KOBAS software. Cufflinks was used for transcript assembly, followed by cuffcompare for transcript comparison to discover new transcripts.

## Analysis of Venn diagrams of DEGs and expression profiles

DEGs from the three tender leaves libraries were analyzed via Venn diagrams using online software (http://bioinformatics.psb.ugent.be/webtools/Venn/). Hierarchical clustering heatmaps and cluster analysis were generated with MultiExperiment Viewer (http://www2.heatmapper.ca/expression/) using the FPKM of selected genes.

## Real-time qPCR analysis

Quantitative real-time PCR (qRT-PCR) was used to detect gene expression and verify the reliability of RNA-seq data. After treatment with DNase I, the total RNAs extracted from tender leaves were used to synthesize first strand cDNA through a RevertAid First Strand cDNA a Synthesis Kit (Fermentas, Vilnius, Lithuania). Transtart Tip Green qPCR SuperMix (TransGen Biotech, Beijing, China) was used to carry out quantitative PCR. Three biological replicates were analyzed. Primers are listed in S1 Table.

## Statistical analysis

The data for tree-structure indexes, sugar metabolism-related enzyme activities, and phytohormone contents in different groups were analyzed using Excel and DPS software (Zhejiang University, China).

# Results

## Growth phenotype comparison of apple trees grafted on different rootstocks

Several key growth indicators of the tree-structure indexes were measured in the fourth year after grafting (Table 1). The diameter at 20 cm above the grafting joint, stem length, number of branches per tree and total branch length per tree values were higher for the apple trees grafted

**Table 1. Statistics of tree-structure indexes in the fifth year after grafting.**

| Index | VR | DIR | DSR |
|---|---|---|---|
| Diameter at 20 cm above grafting joint (cm) | 37.77 ± 1.37a | 32.70 ± 0.55b | 20.61 ± 0.62c |
| Stem length (cm) | 277.33 ± 12.77a | 246.00 ± 7.00b | 182.83 ± 4.76c |
| Number of branches per tree (cm) | 22.00 ± 2.89a | 18.00 ± 1.15a | 9.67 ± 1.86b |
| Total branch length per tree (cm) | 1706.00 ± 234.89a | 1540.50 ± 69.69a | 636.83 ± 96.40b |
| Average branch length per plant (cm) | 77.42 ± 0.59ab | 85.84 ± 2.56a | 69.60 ± 12.27b |

The data represent the means of ten independent experiments. The letters next to the values demonstrate significant difference at 0.05 thresholds.

on VR than for trees grafted on DIR and DSR. The apple trees grafted on DIR showed the longest average branch length per plant. The apple trees grafted on DSR showed the lowest values for all the indexes. The data showed that grafting on different rootstocks changed the growth phenotype of '*Fuji*' apple trees.

## Sugar contents, sugar metabolism-related enzyme activities, and hormone contents in tender leaves and buds

The sucrose content in tender leaves was higher in DIR trees than in the other trees (Fig 1A), while the sucrose content in buds was slightly higher in VR trees than in the other trees (Fig 1B). The glucose content in tender leaves and buds was higher in VR and DIR trees than in

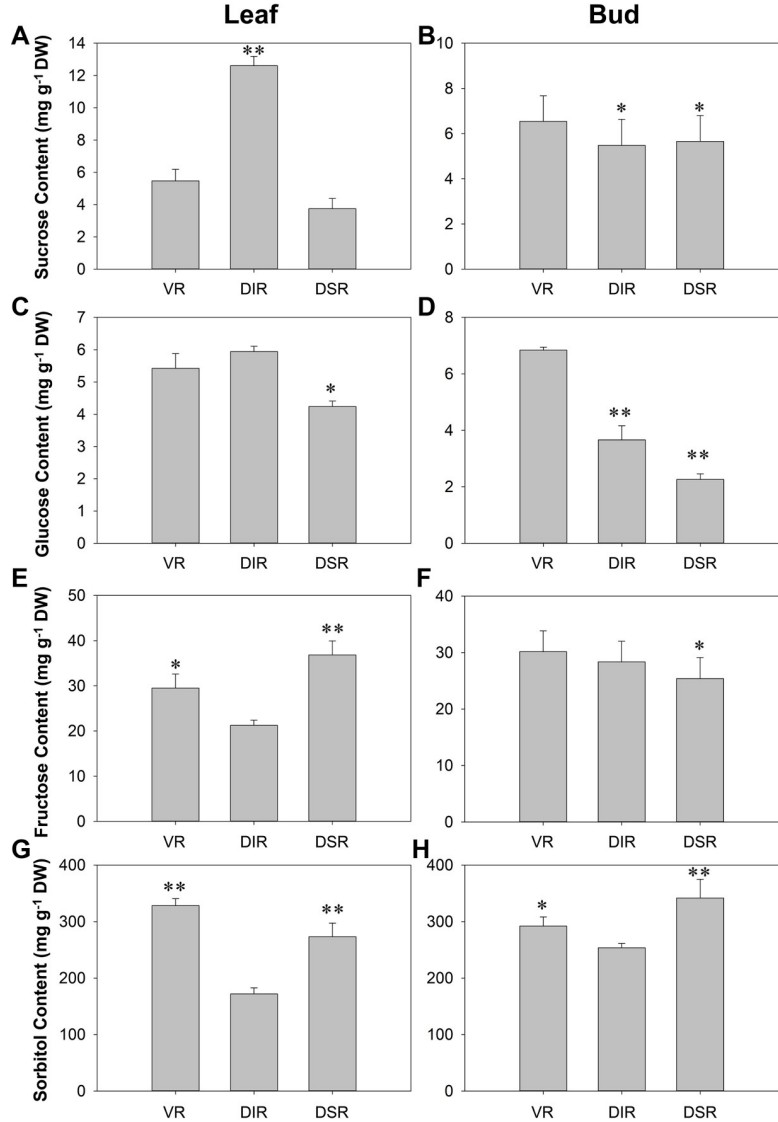

**Fig 1. Sugar content in leaves and buds of 'Fuji' apple grafted on different rootstocks at the capping period of spring shoots.** VR: Vigorous rootstock; DIR: Dwarfing inter-stock; DSR: Dwarfing self-rootstock. (A-B) Sucrose content; (C-D) Glucose content; (E-F) Fructose content; (G-H) Sorbitol content. Values are the means of three replicates ± SE. * and ** indicate significant differences at P < 0.05 and P < 0.01 according to Duncan's multiple range test.

DSR trees (Fig 1C and 1D). The fructose content in tender leaves was highest in DSR trees, followed by VR trees, with DIR trees exhibiting the lowest level (Fig 1E). There were no significant differences in fructose content in buds among all the trees (Fig 1F). The sorbitol content in tender leaves from VR trees was obviously higher than that in leaves from DIR and DSR trees (Fig 1G), and the sorbitol content in buds from DSR trees was higher than that in buds from other trees (Fig 1H).

The differences between SS (sucrose synthase) activities in tender leaves of apple trees grafted on different rootstocks were slight (Fig 2A). While the SS activity in buds from DSR trees was obviously lower than that in buds from other trees (Fig 2B). The SPS (sucrose phosphate synthase) activity in tender leaves from VR trees was lower than that in leaves from other trees, whereas the activity in buds was higher than that in leaves (Fig 2C and 2D). The SDH (sorbitol dehydrogenase) activity in tender leaves and buds from the three types of apple trees showed the same trend. The DIR trees showed lower SDH activity both in tender leaves and buds (Fig 2E and 2F). The AI (acid invertase) activity in both tender leaves and buds from VR trees was significantly higher than in DIR and DSR trees (Fig 2G and 2H). However, the NI (neutral invertase) activity in tender leaves and buds from VR trees was significantly lower than in buds from DIR and DSR trees (Fig 2I and 2J).

The IAA content in tender leaves of apple trees grafted on VR, DIR and DSR was decreased in turn (Fig 3A). The buds of DIR trees had the highest IAA content among the trees, and the IAA content was similar in the buds of VR and DSR trees (Fig 3B). The CK content in the leaves of DSR trees was lower than that in the other trees, while the buds of VR trees showed the lowest level (Fig 3C and 3D). The GA content in tender leaves and buds of VR, DIR, and DSR trees was decreased in turn (Fig 3E and 3F). In the tender leaves and buds of DSR trees, the ABA content was relatively higher. The ABA content in leaves of VR trees was slightly higher than that in DIR trees, while that in the buds of VR trees was lower than levels observed in DIR trees (Fig 3G and 3H).

## Analysis of RNA-seq data

To resolve the mechanism affecting the tree growth phenotype after grafting on different rootstocks, tender leaves adjacent to the top buds of VR, DIR and DSR trees were used to construct RNA-seq libraries. Detailed information regarding the sequencing data is presented in S2 Table. The average total reads of 45,752,279, 43,079,401 and 42,813,045 were generated for VR, DIR and DSR, respectively, and the mapped ratios for the percentage of clean reads located in the reference genome in all clean reads were all about 90% (S2 Table). A total of 3610 DEGs were identified through pairwise comparisons, and the statistical data for comparisons of DEGs among the three samples are displayed in Fig 4A and S3 Table. The general overview of the expression pattern is visualized in a heat map, which provides an overall understanding of the changes in gene expression (Fig 4B). The expression patterns of most DEGs were opposite between VR and DIR trees, and the expression patterns of DEGs were relative similar between DIR and DSR trees. GO classification and KEGG pathway enrichment analysis results for the annotated DEGs in pairwise comparisons of the three groups are listed in Figs S1 and 5, respectively. The 'Glycolysis/ Gluconeogenesis', 'Starch and sucrose metabolism', and 'Plant hormone signal transduction' pathways were all significantly enriched in pairwise comparisons. Furthermore, 1293 new genes were identified and were subjected to GO classification and KEGG analysis (S2 Fig).

## Expression profile of sugar metabolism-related genes in tender leaves

Cluster analysis of transcriptome data showed that sugar metabolism-related gene (a total of 129 genes) expression patterns could be divided into five clusters (Fig 6 and S4 Table). Genes

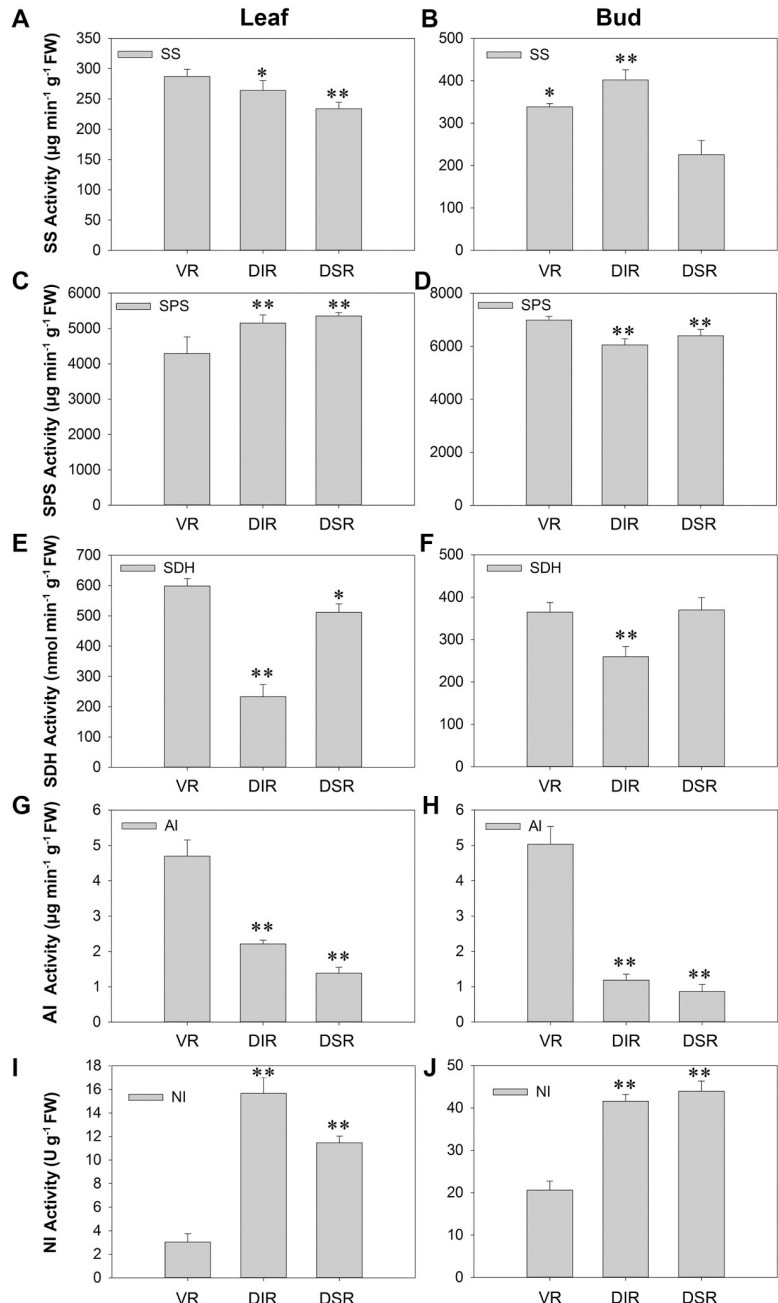

**Fig 2. Activities of enzymes associated with sugar metabolism in leaves and buds of 'Fuji' apple grafted on different rootstocks.** VR: Vigorous rootstock; DIR: Dwarfing inter-stock; DSR: Dwarfing self-rootstock. (A-B) SS, sucrose synthase; (C-D) SPS, sucrose phosphate synthase; (E-F) SDH, sorbitol dehydrogenase; (G-H) AI, soluble acid invertase; (I-J) NI, neutral invertase. Values are the means of three replicates ± SE. * and ** indicate significant differences at $P < 0.05$ and $P < 0.01$ according to Duncan's multiple range test.

expression levels in cluster 1 (13 genes) were higher in the tender leaves of DIR and DSR trees than in VR trees. For example, sucrose synthase (*SUS4*, *SUS5*, and *SUS6L*) and galactinol synthase 1-like (*GolS1-like*) genes showed higher transcript accumulation in leaves of DIR and DSR trees. For cluster 2 (45 genes), the gene expression levels were all high, with slight differences. In this cluster, the sucrose-phosphate synthase 4 (*SPS4*) and sucrose synthase 5-like

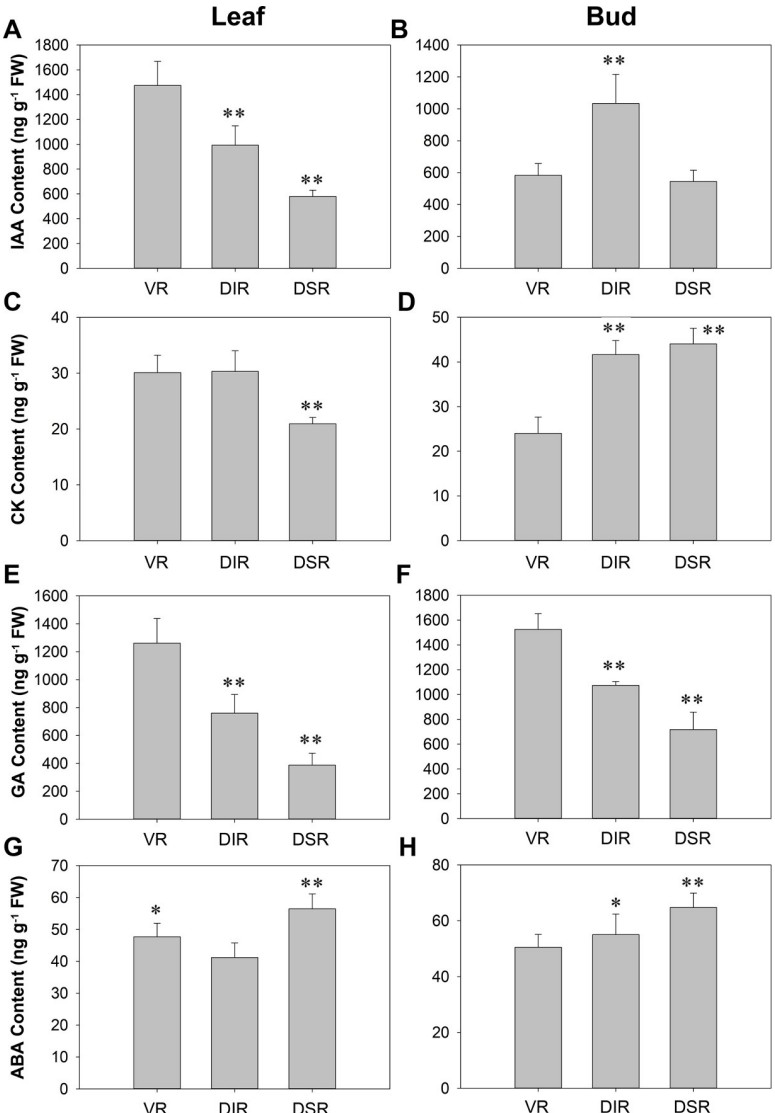

**Fig 3. Hormone contents in leaves and buds of 'Fuji' apple grafted on different rootstocks at the capping period of spring shoots.** VR: Vigorous rootstock; DIR: Dwarfing inter-stock; DSR: Dwarfing self-rootstock. (A-B) IAA, Auxin; (C-D) CK, cytokinin; (E-F) GA, gibberellin; (G-H) ABA, abscisic acid. Values are the means of three replicates ± SE. * and ** indicate significant differences at P < 0.05 and P < 0.01 according to Duncan's multiple range test.

(*SS5-LIKE*) genes are involved in sucrose biosynthesis, the alpha-glucosidase (*AGLA*) and glucan endo-1,3-beta-glucosidase-like (*GIL*) genes are involved in glucose metabolism, and the NADP-dependent D-sorbitol-6-phosphate dehydrogenase-like (*S6PDHL*) gene can promote sorbitol accumulation. The gene expression levels in cluster 3 (11 genes) were highest in the leaves of VR trees. In this cluster, beta-galactosidase (*β-GAL*), which is associated with neutral sugars, exhibited a higher expression level in the leaves of VR trees than in the other trees. There were no obvious differences in the gene expression levels in cluster 4 (55 genes); all genes in this cluster exhibited a low expression level. The genes in cluster 5 (5 genes), such as 7-deoxyloganetin glucosyltransferase-like (*UGTL*), showed the highest expression level in the leaves of DSR trees.

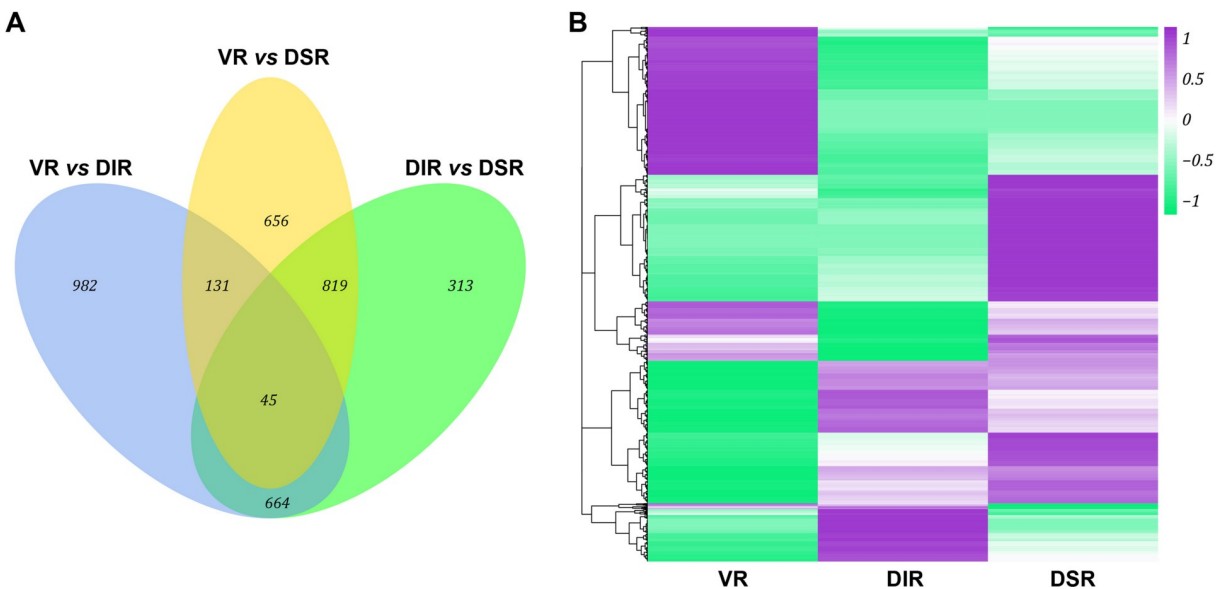

**Fig 4. Transcriptome changes in 'Fuji' apple grafted on three different rootstocks.** (A) Number of DEGs in pairwise comparisons of the three treatments; (B) Heat map diagrams showing the relative expression levels of total DEGs among the three treatments.

### Expression profile of phytohormone-related genes in tender leaves

There were two clusters of phytohormone-related gene expression patterns (Fig 7 and S5 Table). In cluster 1 (7 genes), the expression levels of genes, such as ABA 8'-hydroxylase (*ABA8OX4*, *ABA8OX4-LIKE*), were higher in DIR trees than in other trees. The expression levels of genes in cluster 2 (9 genes) were lower in the tender leaves of VR trees than in the other trees. For example, *RGL2-2*, a member of the GRAS family that is involved in GA signal transduction pathways, and auxin efflux carrier components (*PIN3* and *PIN8*) all showed significantly higher transcript accumulation in leaves of DIR and DSR trees.

### Expression profile of transcription factors in tender leaves

Cluster analysis of transcriptome data revealed that the expression patterns of transcription factors could be divided into two clusters (Fig 8 and S6 Table). For cluster 1 (14 genes), the expression levels of genes, such as ethylene-responsive transcription factor (*ERF034-like*), auxin response factor (*ARF7-LIKE*), and zinc finger protein *CONSTANS-LIKE* (*COL4*), were highest in the leaves of VR trees. For cluster 2 (14 genes), the gene expression levels were all relatively high in each group, and levels in the VR trees were slightly lower than in the other trees. *SHOOT MERISTEMLESS-like* (*STL*), ethylene-responsive transcription factor (*ERF105*) and transcription factor (*MYC2* and *MYC2-like*) showed higher transcript accumulation in the tender leaves of DIR and DSR trees.

### Verification of gene expression via real-time PCR

To confirm the RNA-Seq analysis, the expression levels of 18 selected candidate DEGs were determined in real-time PCR assays. The results indicated that the expression patterns of these genes fitted the data determined by RNA-Seq analysis. For instance, *SUS4*, *PIN3* and *MYC2* were all highly abundant transcripts that also exhibited relatively high expression levels in real-time PCR analysis. These results indicate that the experimental data analysis process was reasonable and that the results were reliable.

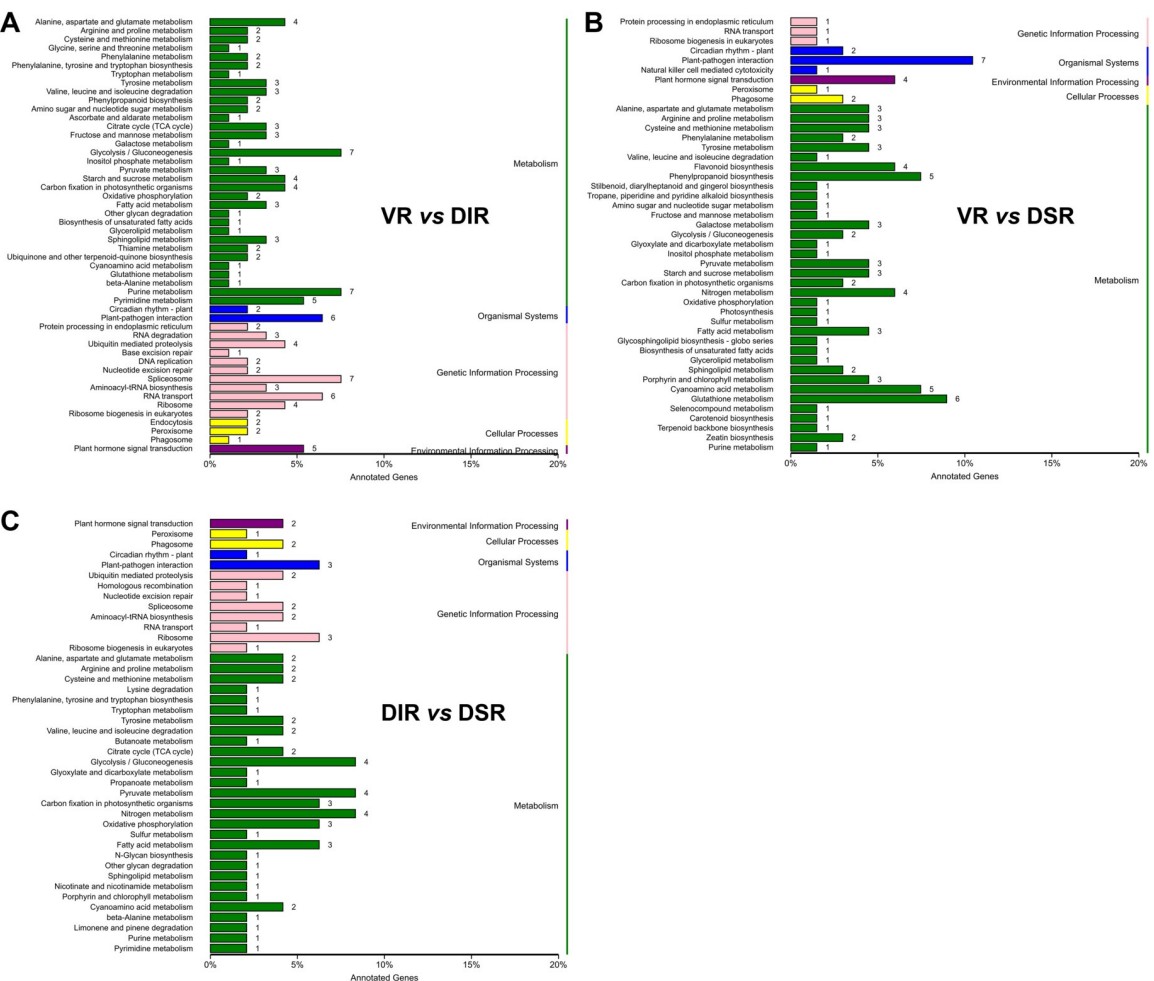

**Fig 5. KEGG pathway enrichment analysis of the annotated DEGs.** DEGs were associated with five different functional categories: metabolism, organismal systems, genetic information processing, cellular processes, environmental information processing.

## Discussion

Grafting joins a plant root system and a shoot from a different individual plant and affects uptake and transport of water and nutrients, production and transport of hormones, and long-distance movement of biological macromolecules in trees [31, 32]. Although these processes have implications for both belowground and aboveground functioning, the tree branch growth decrease induced by rootstock is needed for modern orchards [33, 34]. In this study, we confirmed that grafting on different rootstocks changed the tree-structure indexes (Table 1). DIR can also induce trees to present a dwarfing growth trend, except in the longest branch length (Table 1). Zhou et al. compared the tree growth parameters among eight apple scion-rootstock combinations and found that the DIR combination '*Red Fuji*'/SH.6/Baleng achieved lower plant height, trunk cross-section area, canopy width, and internode length of the shoots than '*Red Fuji*' grafted onto the VR Baleng [2]. The consistency of these experimental results also verifies that the rootstock groups we sought to compare were reasonable. It is likely that the scion effects on rootstocks are ubiquitous due to the flow of sugar, hormones, and nucleic acids into the vascular tissue system [35].

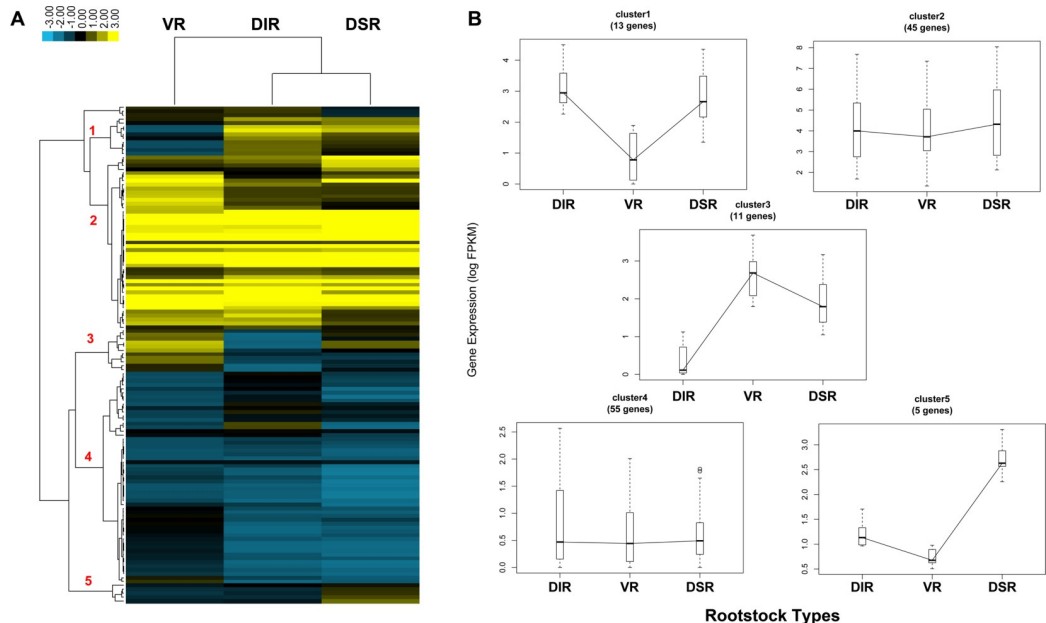

**Fig 6. Expression analysis of genes related to sugar metabolism and signaling identified by RNA-Seq.** (A) Expression profile and cluster analysis of sugar metabolism and signaling was performed with similar expression patterns. Values of FPKM were used for the cluster analysis. Expression data for a given gene are shown relative to its expression in the leaves of apple trees grafting on vigorous rootstock (VR), dwarfing interstock (DIR), and dwarfing self-rootstock (DSR). Red colored numbers indicated on the dendrogram are assigned to major clusters. (B) Trend analysis of representative genes from each of the major clusters 1–5 shown in (A).

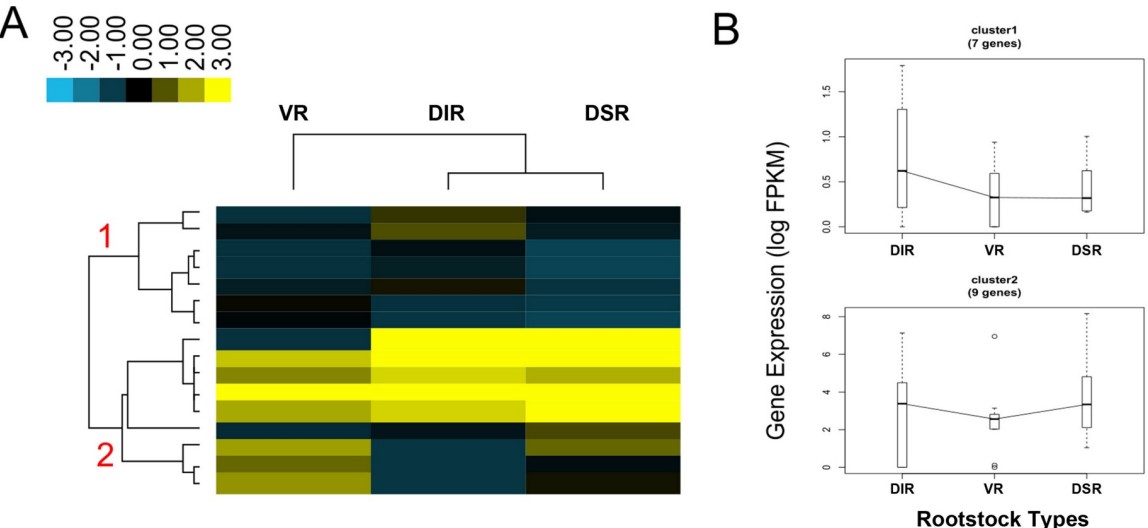

**Fig 7. Expression analysis of genes related to phytohormone metabolism and signaling identified by RNA-Seq.** (A) Expression profile and cluster analysis of phytohormone metabolism and signaling was performed with similar expression patterns. Values of FPKM were used for the cluster analysis. Expression data for a given gene are shown relative to its expression in the leaves of apple trees grafting on vigorous rootstock (VR), dwarfing interstock (DIR), and dwarfing self-rootstock (DSR). Red colored numbers indicated on the dendrogram are assigned to major clusters. (B) Trend analysis of representative genes from each of the major clusters 1–2 shown in (A).

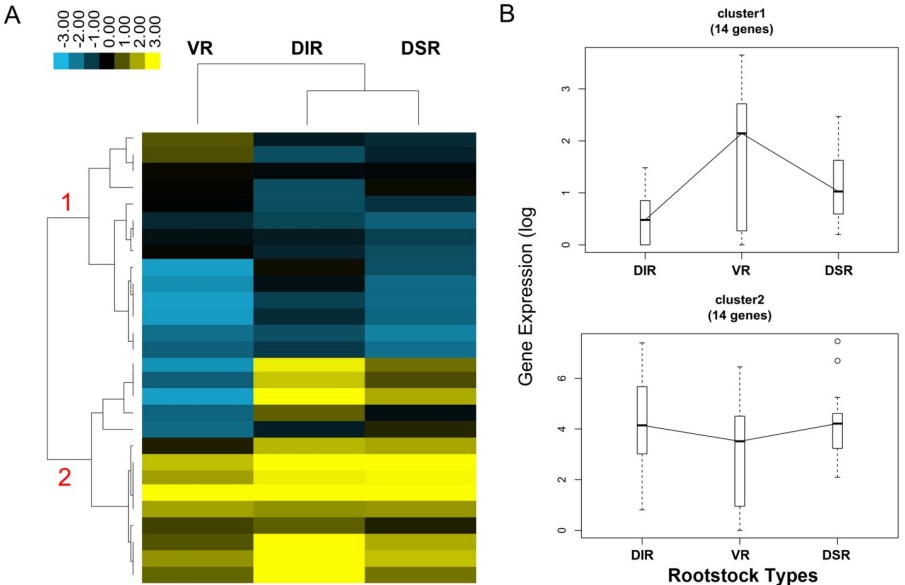

**Fig 8. Expression analysis of transcription factors identified by RNA-Seq.** (A) Expression profile and cluster analysis of transcription factors was performed with similar expression patterns. Values of FPKM were used for the cluster analysis. Expression data for a given gene are shown relative to its expression in the leaves of apple trees grafting on vigorous rootstock (VR), dwarfing interstock (DIR), and dwarfing self-rootstock (DSR). Red colored numbers indicated on the dendrogram are assigned to major clusters. (B) Trend analysis of representative transcription factors from each of the major clusters 1–2 shown in (A).

## Sugar metabolism and signaling aspect

In the leaves of VR trees, the glucose and sorbitol contents were at higher levels. In the leaves of DIR trees, the sucrose and glucose contents remained higher, while in the leaves of DSR trees, the fructose and sorbitol contents were higher. The sugar contents were relatively balanced in the buds of each group, except the glucose content was higher in the buds of VR trees and the sorbitol content was slightly higher in DSR trees (Fig 1). We found that sorbitol was the main sugar in both the leaves and buds (Fig 1G and 1H), and its content difference may be the main factor influencing tree growth. Samuolienė et al. found that in the leaves of the apple cultivar '*Ligol*' grafted onto several types of dwarfing rootstocks, the glucose, fructose and sorbitol contents were also significantly different [36].

In the pathways of sugar metabolism, SS and SPS catalyze the conversion of fructose and fructose-6-phosphate to sucrose, SDH catalyzes the conversion of sorbitol dehydrogenation to fructose, and AI and NI catalyze sucrose decomposition to fructose and glucose [37, 38]. In our study, except NI activity, the activity of all the sugar metabolism-related enzymes was maintained at high levels in both the leaves and buds of VR trees (Fig 2). The activities of these enzymes in leaves and buds of DIR trees presented similar trends. For example, in both the leaves and buds of DIR trees, the activities of SS, SPS and NI were high, while SDH and AI activities were low. Similar conditions were found in the leaves and buds of DSR trees, except the lowest level of SS activity was observed in buds (Fig 2). The SDH activity in these samples followed the same order as the sorbitol content (Figs 1G and 1H, 2E and 2F). This may indicate that sorbitol plays an important role in nutritional synthesis induced by the different rootstocks. The differences in these enzyme activities lead to the differences in sugar contents. As the primary products of photosynthesis in apple are sorbitol and sucrose, these compounds are

transported for export from source leaves to sink tissues [36, 39]. Consequently, the changes in sugar contents may lead to the different inner growth conditions observed in branches.

In the transcriptome data, 128 DEGs related to sugar metabolism were identified and divided into five clusters according to their expression trends (Fig 6 and S4 Table). Sucrose synthases (*SUS4*, *SUS5*, and *SUS6L*) were detected in the leaves of VR, DIR and DSR trees (S4 Table). Different *SUS* genes in the genome of *Malus domestica* (*MdSUSs*) play distinct roles in the sink-source sugar cycle and sugar utilization in different apple sink tissues [40]. Functional studies also suggest that *SUS* genes participate in the shoot apical meristem (SAM) and in early leaf development [41]. Abnormal cotyledons and leaf morphology were exhibited in transgenic tomatoes with suppressed *SlSUS1*, *SlSUS3*, and *SlSUS4* expression, and the expression of auxin-related genes in the SAM and the transport of auxin also altered in these plant lines [42]. The expression levels of galactinol synthase 1-like (*GolS1-like*) were higher in the leaves of dwarfing trees than in VR trees (S4 Table and Fig 9). Galactinol synthase (GolS) controls the rate-limiting step of raffinose-family oligosaccharide (RFO) biosynthesis by synthesizing galactinol from myo-inositol and UDP-galactose, and the accumulation of galactinol and RFOs can protect the photosynthetic apparatus when plants are under oxidative stress [43, 44]. It may be speculated that *GolS1-like* contributes to stress resistance of dwarfing rootstock apples. There were no obvious differences in the high expression levels of *SPS4*, *SS5-LIKE*, *AGLA*, *GIL*, and *S6PDHL* among the groups (S4 Table and Fig 9). These genes promote sucrose biosynthesis, glucose metabolism and sorbitol accumulation to maintain activity in leaves. Although sorbitol contents and SDH activities were found to exhibit coordinated differences in the leaves of each group, the expression levels of *S6PDHL* were similar (Figs 1, 2 and 9). Different transgenic apples transformed with the same *S6PDH* overexpression vector exhibited decreased or increased levels of S6PDH mRNA, protein and activity, and the sorbitol content was not increased in some transgenic lines, while the sucrose content was increased several fold in all lines [45]. Considering the regulatory role of S6PDH in partitioning between sorbitol and sucrose in apple leaves, the inconsistent expression level might be related to the sorbitol content. Beta-galactosidase (β-GAL) removes galactosyl and arabinosyl residues from polysaccharides in the cell wall and hence modifies cell wall expansion [46, 47]. The expression level of *β-GAL* was higher in the leaves of VR trees than in dwarfing trees (S4 Table and Fig 9), and thus *β-GAL* may promote the leaf expansion growth of VR trees. Glycosyltransferases (UGTs) use UDP-sugars, such as UDP-glucose, as glycosyl donors to mediate glycosylation from hormones and secondary metabolites to biotic and abiotic chemicals [48–50]. In the leaves of DSR trees, *UGTL* showed the highest expression level (S4 Table and Fig 9) and thus may accelerate sugar transformation in DSR trees. The complicated expression profiles of sugar metabolism-related genes contribute to the differences in sugar contents and enzyme activities in leaves, and therefore, the tree architecture growth is influenced and changes with grafting on different rootstocks.

## Phytohormone signaling aspect

The IAA, CK, GA, and ABA contents in leaves from VR, DIR and DSR trees were arranged in the order of high to low, except the ABA content was higher in the DSR group (Fig 3). The IAA and CK contents showed the lowest values in the buds of VR trees, while the GA content was the highest. The buds of DIR trees contained the highest IAA content and intermediate contents of the other three hormones. In the buds of DSR trees, the IAA and GA contents were at the lowest levels, while CK and ABA contents were the highest (Fig 3). It is essential that auxins direct the pathway to reconnect the vascular tissue of the new grafting scion and rootstock [51]. It was lower that the content of indole-3-acetic acid (IAA) in dwarfing '*Fuji*'/

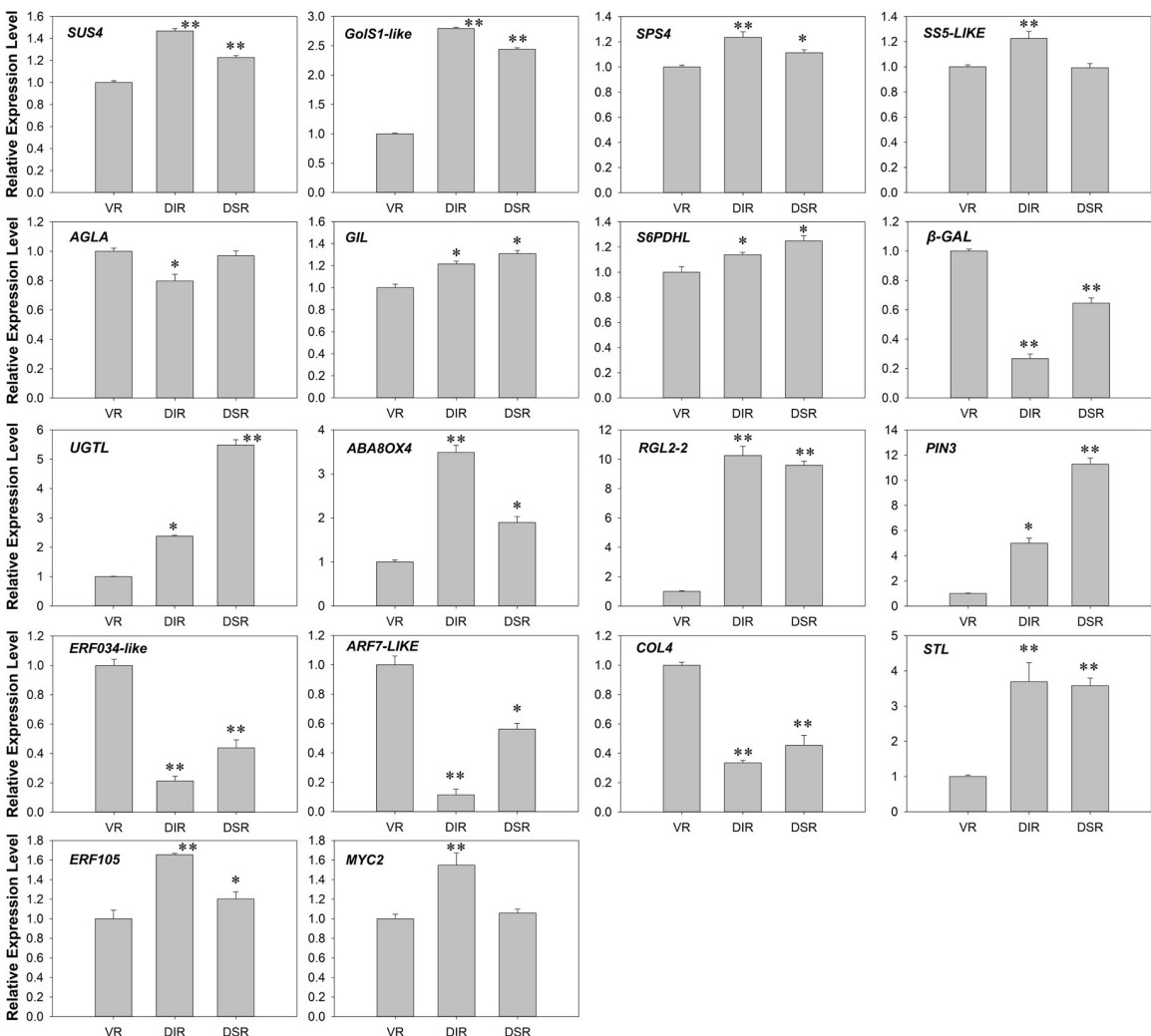

**Fig 9. Real-time PCR verification of gene expression in the tender leaves of 'Fuji' apple grafted on three different rootstocks.** VR: Vigorous rootstock; DIR: Dwarfing inter-stock; DSR: Dwarfing self-rootstock. * and ** indicate significant differences at P < 0.05 and P < 0.01 according to Duncan's multiple range test.

M9 than in the vigorous '*Fuji*'/MM106 [52]. The difference in IAA content in this study was consistent with the viewpoint that auxin concentrations are consistently lower in the cambial region of stems from dwarfing apple rootstock than in stems from more vigorous rootstocks [53]. A decrease in root-produced CK supplied to the scion on a dwarfing apple rootstock was postulated to limit shoot extension growth of the scion, and elevated ABA and reduced GA were associated with dwarfing rootstocks [34, 54]. These previous studies completely support our results showing that grafting on VR, DIR and DSR indeed changed the phytohormones content. After analysis of transcriptome data, 16 DEGs associated with phytohormone signaling were selected (Fig 7 and S5 Table). Several hundreds of hormone-related genes involved in bud development, bag removal, and shoot bending during apple cultivation were analyzed [22, 55, 56]. However, there were not many DEGs related to hormones in this study. This may be due to the relatively short-term experimental treatments carried out in those studies and the sharp change in gene transcription. Here, although different types of rootstocks were grafted, the apple trees had been growing stably for three years, and although differences in the

hormone contents existed, the gene expression levels did not change dramatically. The significantly higher transcript accumulation of auxin efflux carrier components (*PIN3* and *PIN8*) contributed to the lower IAA content in leaves of dwarfing trees (Fig 3 and S5 Table). The expression levels of *ABA8OX4* and *ABA8OX4-LIKE* were highest in the leaves of DIR trees (Fig 9 and S5 Table). As the predominant pathway of ABA catabolism, ABA 8′-hydroxlyase catalyzes the hydroxylation of ABA at the 8′-position to produce 8′-hydroxy ABA, which then spontaneously isomerizes to phaseic acid [57]. The high expression levels of *ABA8OX4* and *ABA8OX4-LIKE* corresponded to the low ABA content in each group, which confirms that *ABA8OXs* play important roles in regulating the ABA content (Figs 3 and 9). *RGL2* is involved in GA signal transduction pathways, while its expression is induced by ABA and its protein is necessary to elevate endogenous ABA, specifically when GA levels are low during *Arabidopsis* seed germination [58]. The expression level of *RGL2-2* was lower in the leaves of VR trees than in leaves of dwarfing trees, while the ABA content was higher in the dwarfing trees, and the GA content was lower (Figs 3 and 9). These findings suggest that *RGL2-2* may also participate in GA and ABA signal transduction in the leaves of apple trees grafted on different rootstocks.

## Regulation role of transcription factors

Two clusters of transcription factors were analyzed in the transcriptome data (Fig 8 and S6 Table). The ethylene-responsive transcription factor *ERF034* was expressed at higher levels in the leaves of VR trees than in leaves of dwarfing trees, while the *ERF105* expression profile was reversed. The *ERF*s represent a large family in *Arabidopsis thaliana* and are involved in the vital biological processes. For example, *AtERF105* enhances the freezing tolerance and cold acclimation of plants [59–61]. In apples, MdERF38 interacts with MdMYB1 to mediate anthocyanin biosynthesis induced by drought stress [62]. The different expression patterns of *ERF034* and *ERF105* may mediate ethylene-induced growth of apple trees grafted on different rootstocks. Auxin response factors (ARFs) are transcription factors that regulate auxin responses in plants, and 31 members have been identified in the apple genome [63, 64]. In tomato fruits, the levels of glucose and fructose became significantly higher in the *SlARF4* downregulated fruit than in the wild-type fruit as the fruit development advanced toward ripening [65]. The expression of *ARF7-LIKE* was higher in the leaves of VR trees, and it may also be involved in sugar metabolism. As an important flowering-time gene, *CONSTANS-LIKE* play roles in the photoperiodic flowering pathway [66]. In the present study, *COL4* showed the highest expression level in the leaves of VR trees, indicating that this gene may play roles in the flowering process, which is an essential stage of tree branch growth. SHOOT MERISTEMLESS (STM) is essential for shoot apical meristem (SAM) function and auxin biosynthesis and transport [67]. In *STM*-overexpressing *Arabidopsis*, the genes involved in the catabolism, signaling and response of auxin, CK, and GA as well as in cell wall modification showed differential expression [68]. *SHOOT MERISTEMLESS-like* (*STL*) was expressed at high levels in the leaves of all three groups in this study, and expression was higher in the leaves of dwarfing trees (S6 Table and Fig 9). Similar expression patterns were also found for the transcription factors *MYC2* and *MYC2-like* (S6 Table and Fig 9). As an important regulator in the jasmonic acid (JA) signaling pathway, in addition to promoting anthocyanin biosynthesis in apple calli, *MdMYC2* also regulates *MdERFs* and ethylene biosynthetic genes to promote ethylene biosynthesis during apple fruit ripening or to reduce aluminum stress tolerance in transgenic plants [69–71]. The different expression pattern of *MYC2* and *MYC2-like* observed in this study indicates that JA signaling is also involved in growth regulation induced by different rootstocks. Further molecular functional studies are needed to verify the regulatory effect of these transcription factors on rootstock-induced growth of apple trees.

## Conclusion

Overall, we have found that grafting on VR, DIR, and DSR changed the branch architecture phenotype of 'Fuji' apple trees by altering the sugar metabolism and hormone signaling in leaves and the expressions of related genes as well as transcription factors. Although the regulation processes are very complex both in physiological or molecular level, and a lot of more detailed and in-depth work can be carried out in the future. The outcomes of this study contribute to an understanding of the responsive molecular interactions between scion-rootstock combinations and have direct applications in apple rootstock breeding.

## Supporting information

**S1 Fig. GO analysis of DEGs in pairwise comparisons of the three groups.**
(TIF)

**S2 Fig. Functional analysis of DEGs identified via transcriptome assembly by Novogene.**
(A) GO enrichment analysis; (B) KEGG enrichment analysis.
(TIF)

**S1 Table. List of primers used for real-time PCR.**
(DOCX)

**S2 Table. Statistics of the DEG numbers among different groups.**
(DOCX)

**S3 Table. Summary of the sequencing data in each sample.**
(DOCX)

**S4 Table. Information of selected sugar-related genes in leaves of apple trees on different rootstocks.**
(DOCX)

**S5 Table. Information of selected phytohormone related genes in leaves of apple trees on different rootstocks.**
(DOCX)

**S6 Table. Information of selected transcription factors in leaves of apple trees on different rootstocks.**
(DOCX)

## Acknowledgments

The authors are very grateful for the constructive comments of the two anonymous reviewers.

## Author Contributions

**Conceptualization:** Yanhui Chen, Xiuhong An, Cungang Cheng.

**Data curation:** Yanhui Chen, Renpeng Ma, Zhuang Li.

**Formal analysis:** Yanhui Chen.

**Funding acquisition:** Cungang Cheng.

**Investigation:** Yanhui Chen, Deying Zhao, Enmao Li, Renpeng Ma.

**Methodology:** Yanhui Chen, Deying Zhao, Enmao Li.

**Resources:** Zhuang Li.

**Writing – original draft:** Yanhui Chen.

**Writing – review & editing:** Yanhui Chen, Cungang Cheng.

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
