## [Decision Letter · Decision Letter 0]

21 May 2020

PONE-D-20-09821

Transcription Profiles Reveal Sugar and Hormone Signaling Pathways Mediating Tree Branch Architecture in Apple (Malus domestica Borkh.) Grafted On Different Rootstocks

PLOS ONE

Dear Dr. Cheng,

Thank you for submitting your manuscript to PLOS ONE. After careful consideration, we feel that it has merit but does not fully meet PLOS ONE’s publication criteria as it currently stands. Therefore, we invite you to submit a revised version of the manuscript that addresses the points raised during the review process.

We would appreciate receiving your revised manuscript by Jul 05 2020 11:59PM. To enhance the reproducibility of your results, we recommend that if applicable you deposit your laboratory protocols in protocols.io, where a protocol can be assigned its own identifier (DOI) such that it can be cited independently in the future. For instructions see: http://journals.plos.org/plosone/s/submission-guidelines#loc-laboratory-protocols

We look forward to receiving your revised manuscript.

Kind regards,

Haitao Shi

Academic Editor

PLOS ONE

Journal Requirements:

https://link.springer.com/article/10.1007/s00344-019-09933-w

https://www.frontiersin.org/articles/10.3389/fpls.2017.00654/full

https://www.cell.com/trends/plant-science/fulltext/S1360-1385(15)00288-5?_returnURL=https%3A%2F%2Flinkinghub.elsevier.com%2Fretrieve%2Fpii%2FS1360138515002885%3Fshowall%3Dtrue

In your revision ensure you cite all your sources (including your own works), and quote or rephrase any duplicated text outside the methods section. Further consideration is dependent on these concerns being addressed.

Reviewers' comments:

Reviewer's Responses to Questions

**Comments to the Author**

1. Is the manuscript technically sound, and do the data support the conclusions?

Reviewer #1: No

Reviewer #2: Partly

2. Has the statistical analysis been performed appropriately and rigorously? 

Reviewer #1: No

Reviewer #2: Yes

3. Have the authors made all data underlying the findings in their manuscript fully available?

Reviewer #1: Yes

Reviewer #2: Yes

4. Is the manuscript presented in an intelligible fashion and written in standard English?

Reviewer #1: No

Reviewer #2: No

5. Review Comments to the Author

Reviewer #1: Transcription Profiles Reveal Sugar and Hormone Signaling Pathways Mediating Tree Branch Architecture in Apple (Malus domestica Borkh.) Grafted On Different Rootstocks

The article by Chen et al. shown the branch growth phenotypes of three combinations of ‘Fuji’ apple trees grafted on different rootstocks (VR; DIR; DSR) were investigated through transcriptomic analysis. The study is interesting and will add more reference datasets to the currently available information. The approach to understand this into plant production could be helpful at large scale. In addition, the authors have performed several other analyses such as phytohomonal analysis through HPLC, and sugar related analytics. However, in current form the manuscript is suffering from severe issues of language, grammar, and typos.

The rational in the introduction creates further lack of clarity of what the authors wants to achieve from this study. In addition to that the data represented in figure format lacks any statistical support. There must be in depth statistical analysis at least one-way ANOVA or t-test to identify significance.

Furthermore, the most essential part is transcriptomics. The authors have not mentioned that how many real biological replicates were used to perform this work. The author must also mention that how many total reads were obtained for each replica or each VR, DIR, DSR. How much this data is correlative to the actual gene content of Apple’s genome.

Furthermore, heatmap for each kind of gene network or regulatory cycle would not help the readers. Whilst keeping in mind the low resolutions of the figures such as 5, 6 and 7. These figures are also not explained properly in the results as well as in figure captions.

In current state, most of gene expression results are explained as high or low, however, there must be p value with DEG files, that is essential to add to the text. Most of the heatmaps are not well explained in the results sections. How many genes were regulated (up or down) for example in case of transcriptional factors or hormones.

Why there are no explanation for certain gene that were exponentially expressed. The validated qRTPCR results are vaguely explain. I think the author should see similar papers to see how this made reader friendly.

The discussion part is weaker and need more emphasis on the results. Further, the methods should be rewritten to explicitly to allow reader to reproduce

Reviewer #2: Line 28: rephrase the sentence. What are the many factors ? such as !!

Lines 33-40: Rearrange the information, with proper citation for the facts declared in the statements. Poorly written and not explained well. Authors are trying to compare different rootstocks in their physiological traits, but not in proper order.

Overall, entire introduction section is not conveying the message leading the rationale for this study. The authors should delineate the studies in most recent literature, and, co-relate to their present studies using the root stocks studied.

Methods:

Line 90-92: It’s not clear about the replicates used, whether 10 or 50 ? How many biological replicates and technical replicates used ?

Line 99: Use ‘briefly’ and replace in subsequent places of the MS.

Results section was well described.

Discussion: Discussion needs to be improved and the entire content be rephrased.

Line 284-286: Authors could include the figures to show differences in length of grafted plants of all rootstocks, although the data represented in table.

Line 297-300: Does not give any novel information.

Line 300-302: Provide meta-analysis of the data with the support of most recent studies to show how the co-relation can contribute.

Overall discussion is very verbose and loaded with too much of basic information. The authors need to focus on the results observed in this study and directly discuss the relationship between the RNA seq data and the metabolites liking to the pathways. I suggest, authors divide this section in multiple subsections and discuss individual pathways affected among the various grafts.

Conclusions:

Line 432-434: This sentence is redundant. The conclusions are repetitive and superfluous. Please revise and provide the novel information about the research findings made in this study.

6. PLOS authors have the option to publish the peer review history of their article (what does this mean?). If published, this will include your full peer review and any attached files.

Reviewer #1: No

Reviewer #2: Yes: Ramesh katam

---

## [Author Response · Author response to Decision Letter 0]

5 Jul 2020

PONE-D-20-09821

Transcription profiles reveal sugar and hormone signaling pathways mediating tree branch architecture in apple (Malus domestica Borkh.) grafted on different rootstocks

Reviewers' comments:

Reviewer #1: The article by Chen et al. shown the branch growth phenotypes of three combinations of ‘Fuji’ apple trees grafted on different rootstocks (VR; DIR; DSR) were investigated through transcriptomic analysis. The study is interesting and will add more reference data sets to the currently available information. The approach to understand this into plant production could be helpful at large scale. In addition, the authors have performed several other analyses such as phytohomonal analysis through HPLC, and sugar related analytics. However, in current form the manuscript is suffering from severe issues of language, grammar, and typos.

(1) The rational in the introduction creates further lack of clarity of what the authors wants to achieve from this study.

Response: Thanks for your kind suggestion. In this study, we intended to explore the molecular determinants of tree branch architecture growth regulation induced by vigorous rootstock (VR), dwarfing interstock (DIR), and dwarfing self-rootstock (DSR). We have reedited the introduction in the revised version (Lines 27-78).

(2) In addition to that the data represented in figure format lacks any statistical support. There must be in depth statistical analysis at least one-way ANOVA or t-test to identify significance.

Response: Yes, we have added the statistical analysis in Figures 1, 2, 3, and 9 in the revised version.

(3) Furthermore, the most essential part is transcriptomics. The authors have not mentioned that how many real biological replicates were used to perform this work. The author must also mention that how many total reads were obtained for each replica or each VR, DIR, DSR. How much this data is correlative to the actual gene content of Apple’s genome.

Response: Yes, we have explained the repetition of the experimental setup in the revised version (Lines 87-91). And we have explained two other issues in detail in the revised version (Lines 206-209).

(4) Furthermore, heatmap for each kind of gene network or regulatory cycle would not help the readers. Whilst keeping in mind the low resolutions of the figures such as 5, 6 and 7. These figures are also not explained properly in the results as well as in figure captions.

Response: Yes, the resolution of the pictures we upload is 600 dpi. It may be that the PDF generated by the system causes the picture to become blurred. There may be an option to view the original image on the peer review system. You can find it. Thank you very much for your kind understanding. In addition, we have added more explanation for these figures in the revised version.

(5) In current state, most of gene expression results are explained as high or low, however, there must be value with DEG files, that is essential to add to the text. Most of the heatmaps are not well explained in the results sections. How many genes were regulated (up or down) for example in case of transcriptional factors or hormones.

Response: Yes, the information of selected DEGs in leaves of apple trees on different rootstocks were shown in supporting Tables 4-6. And we have reedited the results sections in the revised version.

(6) Why there are no explanation for certain gene that were exponentially expressed. The validated qRTPCR results are vaguely explain. I think the author should see similar papers to see how this made reader friendly.

Response: Thanks for your kind advice. In our sequencing data, we did not find genes whose expression was up-regulated or down-regulated sharply, probably because the overall impact of grafting different rootstocks on fruit trees was a long-term slow process rather than a rapid process. We used quantitative PCR to verify the reliability of transcriptome sequencing data. The genes analyzed in Figure 9 are important functional genes and transcription factors discussed in the results section. Therefore, in the interpretation of Figure 9, we choose to use the form of simplified description.

(7) The discussion part is weaker and need more emphasis on the results. Further, the methods should be rewritten to explicitly to allow reader to reproduce.

Response: Yes, we have reorganized these contents in the revised version.

Reviewer #2:

(1) Line 28: rephrase the sentence. What are the many factors ? such as !!

Response: Thank you for your kind suggestion. We have added the factors in the revised version (Lines 28-30).

(2) Lines 33-40: Rearrange the information, with proper citation for the facts declared in the statements. Poorly written and not explained well. Authors are trying to compare different rootstocks in their physiological traits, but not in proper order.

Response: Yes, we have rewritten these sentences in the revised version (Lines 33-39).

(3) Overall, entire introduction section is not conveying the message leading the rationale for this study. The authors should delineate the studies in most recent literature, and, co-relate to their present studies using the root stocks studied.

Response: Thanks for your kind advice. We have recited several recent references in the introduction section in the revised version. And we have delineated the reasons for why we studied Malus baccata and T337 rootstocks in the revised version (Lines 59-61, 75-76 ).

(4) Line 90-92: It’s not clear about the replicates used, whether 10 or 50? How many biological replicates and technical replicates used ?

Response: Thanks. We have explained the repetition of the experimental setup in the revised version (Lines 87-91).

(5) Line 99: Use ‘briefly’ and replace in subsequent places of the MS.

Response: Yes, we have replaced them in the revised version (Lines 96 and 107).

(6) Results section was well described.

Response: Thank you very much for your affirmation. Your comments have benefited us a lot.

(7) Discussion: Discussion needs to be improved and the entire content be rephrased.

Response: Thanks for your constructive suggestions. We have divided the discussion into three sections and rearranged the contents in the revised version (Lines 289-428).

(8) Line 284-286: Authors could include the figures to show differences in length of grafted plants of all rootstocks, although the data represented in table.

Response: Thanks for your advice. Figures could show the differences more clearly, but we did not save appropriate photos when sampling. So we can only use statistical data to explain the differences in tree growth. We are very sorry and hope to get your understanding.

(9) Line 297-300: Does not give any novel information.

Response: Yes, we have deleted these repeated sentences in the revised version.

(10) Line 300-302: Provide meta-analysis of the data with the support of most recent studies to show how the co-relation can contribute.

Response: We are sorry that this sentence was wrong. Our intention at that time was to say that the clustering of ‘glucose metabolism’ and ‘hormone signaling’ pathways was also significantly enriched in Figure 5. We are very sorry for the inconvenience caused by our mistake. Considering that these sentences are repetitive descriptions, we have deleted them in the revised version. If you feel that this modification is incorrect, we are very willing to revise it again.

(11) Overall discussion is very verbose and loaded with too much of basic information. The authors need to focus on the results observed in this study and directly discuss the relationship between the RNA seq data and the metabolites liking to the pathways. I suggest, authors divide this section in multiple subsections and discuss individual pathways affected among the various grafts.

Response: Yes, thanks for your constructive suggestions. We have divided the discussion into three sections and have deleted verbose contents in the revised version (Lines 289-428).

(12) Conclusions: Line 432-434: This sentence is redundant. The conclusions are repetitive and superfluous. Please revise and provide the novel information about the research findings made in this study.

Response: Yes, we have deleted these lines and reedited the conclusions in the revised version (Lines 429-436).

---

## [Editor Report · Decision Letter 1]

9 Jul 2020

Transcription profiles reveal sugar and hormone signaling pathways mediating tree branch architecture in apple (Malus domestica Borkh.) grafted on different rootstocks

PONE-D-20-09821R1

Dear Dr. Cheng,

We’re pleased to inform you that your manuscript has been judged scientifically suitable for publication and will be formally accepted for publication once it meets all outstanding technical requirements.

Kind regards,

Haitao Shi

Academic Editor

PLOS ONE

---

## [Editor Report · Acceptance letter]

13 Jul 2020

PONE-D-20-09821R1 

Transcription profiles reveal sugar and hormone signaling pathways mediating tree branch architecture in apple (*Malus domestica* Borkh.) grafted on different rootstocks 

Dear Dr. Cheng:

I'm pleased to inform you that your manuscript has been deemed suitable for publication in PLOS ONE. Congratulations! Your manuscript is now with our production department. 

Kind regards, 

on behalf of

Dr. Haitao Shi 

Academic Editor

PLOS ONE